# Oxidative Stress and Antioxidant Therapies in Friedreich’s Ataxia

**DOI:** 10.3390/cells14181406

**Published:** 2025-09-09

**Authors:** Félix Javier Jiménez-Jiménez, Hortensia Alonso-Navarro, Elena García-Martín, Alba Cárcamo-Fonfría, Miguel Angel Martín-Gómez, José A. G. Agúndez

**Affiliations:** 1Section of Neurology, Hospital Universitario del Sureste, Arganda del Rey, E 28500 Madrid, Spain; hortalon@yahoo.es (H.A.-N.); alba.carcamo@salud.madrid.org (A.C.-F.); mmartingomez7@salud.madrid.org (M.A.M.-G.); 2University Institute of Molecular Pathology Biomarkers, Universidad de Extremadura, E 10071 Caceres, Spain; elenag@unex.es (E.G.-M.); jagundez@unex.es (J.A.G.A.)

**Keywords:** Friedreich’s ataxia, pathogenesis, oxidative stress, biological markers, frataxin, animal models

## Abstract

The pathogenesis of Friedreich’s ataxia (FRDA) remains poorly understood. The most important event is the deficiency of frataxin, a protein related to iron metabolism and, therefore, involved in oxidative stress. Studies on oxidative stress markers and gene expression in FRDA patients have yielded inconclusive results. This is largely due to the limited number of studies, small sample sizes, and methodological differences. A notable finding is the decreased activity of mitochondrial respiratory chain complexes I, II, and III, as well as aconitase, in endomyocardial tissue. In contrast, numerous studies in experimental models of FRDA (characterized by frataxin deficiency) have shown evidence of the involvement of oxidative stress in cellular degeneration. These findings include increased iron concentration, mitochondrial dysfunction (with reduced respiratory chain complex activity and membrane potential), and decreased aconitase activity. Additionally, there is the induction of antioxidant enzymes, reduced glutathione levels, elevated markers of lipoperoxidation, and DNA and carbonyl protein oxidation. The expression of NRF2 is decreased, along with the downregulation of PGC-1α. Therefore, it is plausible that antioxidant treatment may help improve symptoms and slow the progression of FRDA. Among the antioxidant treatments tested in FRDA patients, only omaveloxolone and, to a lesser extent, idebenone (particularly for cardiac hypertrophy) have shown some efficacy. However, many antioxidant drugs have shown the ability to reduce oxidative stress in experimental models of FRDA. Therefore, these drugs may be useful in treating FRDA and are likely candidates for future clinical trials. Future studies investigating oxidative stress and antioxidant therapies in FRDA should adopt a prospective, multicenter, long-term, double-blind design.

## 1. Introduction

Between 1863 and 1877, Nicolas Friedreich described a clinical syndrome in nine patients from five families. The condition began at puberty and included progressive gait and limb ataxia, along with dysarthria. Over time, additional symptoms appeared, such as areflexia, sensory deficits, nystagmus, muscle weakness, scoliosis, diabetes, and tachycardia. In four of the patients, autopsy studies showed degeneration of the posterior funiculus, posterior spinal roots, Clarke’s spine, and lateral funiculus, and three presented evidence of cardiomyopathy [1]. Therefore, Friedreich is considered to be the first author to describe the concept of hereditary ataxia and to carry out a clinical–pathological study of a form of spinocerebellar degeneration. 

Friedreich’s ataxia (FRDA), with a prevalence of approximately 0.5–1 per 100,000, is the most common hereditary recessive ataxia. Clinically, FRDA presents with gait and limb ataxia, dysarthria, areflexia, and loss of vibratory and proprioceptive sensation. Additional features include scoliosis (70–75%), pes cavus (55–75%), cardiomyopathy with atrial arrhythmias and heart failure (40–50%), vision loss due to optic neuropathy (40–50%), hearing loss from auditory nerve axonopathy (15–20%), and diabetes mellitus (5–10%) [2].

FRDA is caused by a heterozygous autosomal recessive mutation. This involves a pathogenic expansion of the GAA trinucleotide repeat in intron 1 of the frataxin gene (*FXN*, *FA*, *X25*, *CyaY*, *FARR*, or *FRDA*; chromosome 9q21.11; gene ID 2395; MIM 606829. https://www.ncbi.nlm.nih.gov/gene/2395 (accessed on 30 July 2025). This mutation is found in 98% of cases. [3]. *FXN* is a nuclear gene that encodes a mitochondrial protein belonging to the FXN family, which is implicated in the regulation of iron transport in the mitochondria (likely related to iron–sulfur protein production and storage and iron chaperone activity), although its presence in the cytosol has also been described [4]. The GAA mutation leads to a marked decrease in the synthesis of FXN. FXN is highly expressed in the spinal cord and heart, and, to a lesser extent, in the cerebellum, liver, pancreas, skeletal muscles, and cerebral cortex [4].

The primary histopathological hallmark of FRDA is the degeneration of dorsal root ganglia. This leads to the secondary degeneration of the dorsal spinal roots, posterior spinal cord columns, gracilis and cuneate nuclei, spinocerebellar tracts, dentate nuclei of the cerebellum, corticospinal tracts, and the heart [4,5]. Although the pathogenic mechanism of FRDA is not fully understood, the marked deficiency of FXN and its role in iron metabolism suggest that oxidative stress plays a central role. Oxidative stress refers to an imbalance between the production of reactive oxygen species (ROS) and the body’s antioxidant defenses, leading to the oxidation of lipids, proteins, and DNA.

This narrative review aims to summarize the current evidence on the role of oxidative stress in the pathogenesis of FRDA. These studies cover several areas: oxidative stress marker concentrations in different tissues from patients diagnosed with FRDA (including proteomic, metabolomic, lipidomic, and multi-omic studies), case-control studies on the possible association of genes related to oxidative stress with the risk for FRDA, studies showing the presence of oxidative stress in experimental models of FRDA, and studies addressing the possible efficacy of antioxidant drugs in the treatment of FRDA—both clinical trials in patients with FRDA or in experimental models of this disease. For this purpose, we performed a literature search encompassing the terms “Friedreich’s ataxia” and “oxidative stress” using the PubMed Database from 1966 to 25 July 2025. We analyzed manually, one by one, the 435 references retrieved by the whole search, and selected only those strictly related to the topic, as described above.

## 2. Oxidative Stress Markers in Patients with FRDA

The main results, methods used for biochemical determinations, and participating subjects in each of the studies addressing oxidative stress markers in different tissues (including brain/spinal cord, plasma, serum, whole blood, erythrocytes, lymphocytes/lymphoblasts, fibroblasts, endomyocardial and skeletal muscle tissues, and urine) from patients with FRDA compared to controls are described in detail in Appendix A [6,7,8,9,10,11,12,13,14,15,16,17,18,19,20,21,22,23,24,25,26,27,28,29,30,31,32,33,34,35,36,37,38] and summarized in Table 1.

Markers of lipid peroxidation, mainly malonyldialdehyde (MDA) and/or thiobarbituric acid (TBA) reactive substances (TBARS), have been found to be similar between FRDA patients and healthy controls (HC) in the cerebellum, brainstem, and cortex, although they have been described as having a decreased susceptibility to in vitro lipid peroxidation in FRDA patients [6]. Plasma MDA levels were found to be increased in FRDA patients in a single study [11], and 4-hydroxynonenal (4-HNE) levels increased in another [30]. However, a lipidomic analysis in the erythrocytes of patients with FRDA and controls did not find significant differences in lipid peroxidation in the two study groups [12]. Variations in study outcomes may be attributed to differences in sample size, assay methodologies, and the disease stage of the participants.

Plasma levels of the marker of hydroxyl radical attack dihydroxybenzoic acid (DHBA) [13] and susceptibility to oxidative stress [11] have been reported to be similar in FRDA patients and HC in two studies, while another described a significant increase in the free radical superoxide in FRDA [33]. 8-hydroxy-2′-deoxyguanosine (8-OHdG), a marker of DNA oxidation, is increased in the urine of FRDA patients [13]. Carbonyl proteins have been found to be increased, and total antioxidant capacity (TAC) decreased in patients with FRDA [12]. However, the sample sizes of most of these studies are relatively small.

Studies addressing mitochondrial respiratory chain complex enzymatic activities or expression in the cerebellum [8], dorsal root ganglia [8], or spinal cord [7]; lymphocytes/lymphoblasts [27]; peripheral blood mononuclear cells (PBMC) [29]; platelets [29]; skin fibroblasts [27]; and skeletal muscle [8,27] showed similar values in FRDA patients and HC. However, studies performed with endomyocardial biopsy samples showed a significant decrease in the activity of mitochondrial respiratory chain complexes I, II, and III [8,27], and there was similar activity in complexes IV [8,27] and V [27] in patients with FRDA compared to HC.

Despite the important role of FXN in iron metabolism, concentrations of iron in the cerebellum [8,9], spinal cord [8], dorsal root ganglia [8], peripheral nerves [8], erythrocytes [23], and skeletal muscle [8] have been reported to be similar to that of HC, although iron levels were decreased in plasma [14], PBMC [29], platelets [29], and skin fibroblasts [31,33,34]. Concentrations or activity of the enzyme aconitase (an important iron–sulfur protein) in FRDA patients, in comparison to HC, have been found to be similar in the cerebellum and dorsal root [8], skin fibroblasts [11,27], and skeletal muscle [8,27], while these were reported as decreased in endomyocardial tissue from FRDA patients [8,27]. Concentrations of other iron-related proteins, such as hemoglobin [26], protoporphyrin IX [26], and ferrochelatase [26] in the erythrocytes of FRDA patients, have been reported to be similar to those of controls, but glutathionyl–hemoglobin was increased [22].

Alpha-tocopherol (vitamin E) concentrations were found to be similar in the cerebellum, brainstem, and cortex in both FRDA patients and HC [6]. Two studies described non-significant differences in serum vitamin E levels between FRDA patients and controls [16,17], while two others described decreased serum vitamin E in FRDA patients [18,19]. Once again, the sample size, assay method, and disease stage could be related to the differences in the results of these studies. Serum coenzyme Q_10_ concentrations are decreased [19], and serum uric acid levels increased [20] in FRDA patients.

Glutathione bound to proteins has been found to be increased in the spinal cord [7] and fibroblasts [32] in FRDA patients. Total glutathione levels were similar in the plasma [12], whole blood [22,23], lymphocytes [28], and fibroblasts [23,32] in FRDA patients and HC, except in one study that showed decreased levels of fibroblasts in FRDA patients [31]. However, free glutathione concentrations were found to be decreased in whole blood [22], and the reduced glutathione–oxidized glutathione (GSH–GSSG) ratio led to a decrease in plasma [12], lymphocytes [28], and fibroblasts [32] in FRDA patients.

Glutathione peroxidase (GPx) activity was found to be decreased in the whole blood of FRDA patients [24] but was similar to that of HC in erythrocytes [25] and fibroblasts [33]. Total superoxide dismutase (SOD) and glutathione-S-transferase (GST) activities were increased in erythrocytes [23], total SOD was decreased in fibroblasts [34], manganese-SOD (Mn-SOD or SOD2) activity and mRNA increased [33] or decreased [34] in the fibroblasts of FRDA patients, copper–zinc SOD (Cu–Zn-SOD or SOD1) activity and mRNA decreased [34] or were similar to that of HC [33], and catalase (CAT) activity [33] was similar to that of HC in fibroblasts. Recently, Quatrana et al. [30] reported a significant upregulation of the antioxidant enzymes thioredoxin 1 (TRX1) and glutaredoxin 1 (GLRX1) in FRDA patients compared to HC, which was associated with the activation of the nuclear factor kappa B (NF-κB) p65 subunit (NF-κB p65) in fibroblasts.

One study using proton magnetic resonance spectroscopy (^1^H-MRS) described decreased N-acetylaspartate (NAA neuroaxonal marker) and choline (a membrane marker) and increased mean diffusivity in the FRDA cerebellum [10]. Lodi et al. [36], using ^31^phosphorus magnetic resonance spectroscopy (^31^P-MRS), showed a decreased phosphocreatine–adenosine triphosphate (ATP) ratio in heart and skeletal muscle—suggesting mitochondrial dysfunction—in FRDA patients compared to HC. Finally, Swarup et al. [15] described the downregulation of seven proteins and the upregulation of four proteins in the plasma of FRDA patients—some of these are related to oxidative stress processes.

Castaldo et al. [39] described a significant telomere shortening (oxidative stress being a modulator of this parameter), assessed by real-time polymerase chain reaction quantitative analysis (RT-qPCR) in leukocytes from 37 patients with FRDA compared to 36 age- and sex-matched HC, as well as an inverse correlation between this value and disease duration. Anjomani Virmouni [40], using RT-qPCR, also described telomere shortening in leukocytes from 18 FRDA patients and 12 age- and sex-matched controls, which was not correlated with GAA repeat size, as well as in autopsy cerebellar tissues (age and sex not provided). Further, they found significantly higher (but dysfunctional) telomeres in fibroblasts from seven FRDA patients compared to three HC. Finally, Scarabini et al. [41], in a cohort of 61 FRDA biallelic patients, 29 heterozygous FRDA carriers, and 87 age-matched HC, using RT-qPCR, showed non-significant differences in leukocyte telomere length (LTL) between FRDA patients and HC (although LTL was longer in patients aged less than 35 years and higher in those aged more than 36 years), while FRDA carriers showed longer LTL than HC. In addition, they described a positive correlation of LTL with GAA repeat length and a negative correlation with age, as well as an association between shorter leukocyte telomeres and the presence of cardiomyopathy.

## 3. Genetic Variants of Genes Related to Oxidative Stress in Patients with Friedreich’s Ataxia

Haugen et al. [42], in a microarray analysis involving 28 FRDA patients (M:F 16:12, mean age 13.5 ± 2.3 years, mean disease duration 6.0 ± 3.5 years) and 10 controls (M:F 8:2, mean age 20.3 ± 1.4 years) in peripheral blood samples, found the downregulation of 899 genes and upregulation of 471 genes, including those related to apoptosis signaling, transcription/RNA processing, cell–cell signaling, the cell cycle, the ubiquitin cycle, proteolysis–protein catabolism, response to stimuli, and fatty acid beta-oxidation.

Hayashi et al. [43] investigated the expression of 84 oxidative stress genes in cellular cultures of B-lymphoblasts from patients with FRDA and HC using real-time quantitative polymerase chain reaction (RT-qPCR) followed by Western blot analysis, and found increased mRNA expression of *PDZ* and *LIM domain 1* (*PDLIM1*), *eosinophil peroxidase* (*EPX*), *glutathione peroxidase 2* (*GPX2*), *phosphatidylinositol-3*,*4*,*5-trisphosphate-dependent Rac exchange factor 1* (*PREX1*), as well as decreased mRNA expression for *periredoxin 5* (*PRDX5*), *ring finger protein 7* (*RNF7*), *dual-specificity phosphatase 1* (*DUSP1*), *periredoxin 2* (*PRDX2*), *neutrophil cytosolic factor 2* (*NCF2*), and *surfactant protein D* (*SFTPD*) genes using primary qPCR. Western blot analysis confirmed the changes in mRNA expression of *PDLIM1*, *PRDX5*, *NCF2*, and *SFTPD* genes.

Steinkellner et al. [26], in a study using cDNA-based microarray (IronChip)-containing genes involved in iron and copper metabolism in three FRDA patients and six matched HC, found the upregulation of mRNA for *endothelial PAS domain protein 1* (*EPAS1*) and *amyloid beta (A4) precursor protein* (*APP*), and the downregulation of *FXN*, *tumor necrosis factor*, *alpha-induced protein 3* (*TNFAIP3*), *vascular endothelial growth factor A* (*VEGF*), *Toll-like receptor 4* (*TLR4*), *FBJ murine osteosarcoma viral oncogene homolog* (*FOX*), *N-myc downstream-regulated 1* (*NDRG1*), *selectin P* (*granule membrane protein 140kDa*, *antigen CD62*, *SELP*), *selenoprotein P plasma 1* (*SEPP1*), and *spermatogenesis-associated 1* (*SPAP1*) genes.

Kelly et al. [44], in a study involving 79 FRDA patients and 170 controls, showed an association between the rs5186 single-nucleotide polymorphism (SNP) in the *angiotensin-II type-1 receptor* (*AGTR1*) gene and FRDA, with this SNP also being related to an increase in interventricular septal wall thickness and left ventricular mass (LVM), while other SNPs in the same gene and in *angiotensin-converting enzyme* (*ACE*) and *ACE2* genes showed a lack of association with FRDA.

Singh et al. [45], in a PCR amplification and sequencing analysis of the D-loop, NADH (nicotin–adenin–dinucleotide) dehydrogenase subunits 1-6 (ND1-6), and ATP regions of the mitochondrial genome (comprising 50% of the latter) involving 30 FRDA patients and 62 controls, showed a significantly higher load of mitochondrial variations per individual, and an over-representation of the non-synonymous variation p.L237M in ND2 in FRDA cases compared to controls, this variation being associated with longevity and myocardial infarction.

Bolotta et al. [12], in a study on leukocytes from seven FRDA patients and seven HC using qPCR, found overexpression of *thioredoxin reductase 1* (*TXNRD1*) and *nuclear factor erythroid 2-like 2* (*NRF2*) genes in FRDA patients, while *SOD1*, *SOD2*, *CAT*, *GPX1*, and *GSH S-reductase* (*GSR*) gene expressions were similar in FRDA patients and HC. Tozzi et al. [25] found non-significant differences in the distribution of *GSTP1-1* genotypes and allelic variants between 14 FRDA patients and 21 age- and sex-matched HC.

McMackin et al. [24], in a study on lymphocytes and skin fibroblasts from FRDA patients and histological dorsal rood ganglion from a mouse model of FRDA, described overlapping gene expression changes primarily affecting antioxidant genes related to selenium metabolism and GPx activity (*selenoprotein W1-SEPW1-*, *GPX7*, and *TXNRD1*), ten genes related to the positive regulation of the apoptotic process, and nine genes related to mitochondrial translation and transcription.

Tiano et al. [46], in a comparative microarray study on lymphoblastoid cells from a FRDA patient with GM15850 transfected cells, found ninety-one differentially expressed genes, seven of them downregulated and eighty-four upregulated, the more significant being *chromosome 19 open reading frame 12* (*C19orf12*, downregulated, which causes a form of neurodegeneration with brain iron accumulation) and *HCLS1-associated X-1* (*HAX-1*, upregulated; *HAX-1* mRNA expression and HAX-1 protein levels were correlated with FXN expression). Subsequently, they analyzed FXN and HAX-1 mRNA and protein expression in peripheral blood from 39 FRDA patients and 28 age- and sex-matched controls and described a decrease in these values in FRDA patients and a correlation between them.

Petrillo et al. [23], in an analysis (by using RT-qPCR) of the GSH-related genes *glutamate–cysteine ligase* (*GCL*, which encodes the step-limiting enzyme of the GSH synthesis) and *GSR* (responsible for the re-cycling of GSH from its oxidized form GSSG) and *NRF2* gene expression involving a FRDA patient, his mother and sister (both asymptomatic carriers), his father, and three age- and sex-matched controls, found a significant upregulation of the *GCL* gene in the leukocytes and fibroblasts of the asymptomatic mother and similar expression levels in the proband, the sister, and the father compared to controls, while the expression levels of *GSR* and *NRF2* were significantly higher in the proband, the mother, and the sister, and similar to those of controls and the father. The same group, in a study performed on skin fibroblasts from three FRDA patients compared to three HC using RT-qPCR and Western blotting, showed a significant decrease in mRNA expression and protein levels of *NRF2*, *NAD(P)H quinone oxidoreductase 1* (*NQO1*), *Heme Oxygenase-1* (*HO-1*), and *GCL* in FRDA patients [23].

Finally, Quesada et al. [47] described an overexpression of iron–sulfur cluster genes and other oxidative stress-related genes, active-caspase 3, and other apoptosis-related genes, as well as the upregulation of brain-derived neurotrophic factor (BDNF), neuregulin 1, and miR-132 in periodontal ligament cells from patients with FRDA compared to HC.

Most of these studies, performed with different methods, need to be confirmed by replication studies.

## 4. Oxidative Stress in Experimental Models of FRDA

Several experimental models of FRDA have been developed in animals, including Saccharomyces cerevisiae yeast, Drosophila flies, the nematode Caenorhabditis elegans, and mice with a reduced expression of FXN [48,49].

A yeast gene (*yeast FXN homolog*, *YFH1*) encodes a mitochondrial protein involved in iron homeostasis and respiratory function that has a high sequence similarity with the human FXN [50]. Radisky et al. [51] showed that yeast strains with the absence of the YHF1 protein suffered from mitochondrial damage that was closely related to the concentration of iron, suggesting that the accumulation of iron, leading to oxidative stress, was responsible for this mitochondrial damage. The reintroduction of the FXN homolog reversed these alterations [51].

Based on these data, Wong et al. [52] showed that the incubation of human fibroblasts from FRDA patients with iron or with hydrogen peroxide caused oxidant-induced death, which was partially reversed by iron chelators (deferoxamine), intracellular Ca^2+^ chelators, and apoptosis inhibitors. In contrast, other authors described that oligomycin was able to induce oxidative stress, inducing SOD1 and SOD2, and the loss of mitochondrial membrane iron-containing enzyme activity in fibroblasts from controls but not in fibroblasts from FRDA patients, as well as the lack of induction of SOD activity in the heart of knock-out mice for the *FXN* gene, suggesting that oxidative damage to iron–sulfur clusters was responsible for the mitochondrial deficiency and long-term mitochondrial iron overload in FRDA [53]. Consistent with this finding, Jiralerspong et al. [54] showed a lack of the normal upregulation of the stress defense protein SOD2 and the absence of increased activation of the redox-sensitive factor NF-κB in FRDA fibroblasts exposed to iron.

Bulteau et al. [55] showed, in a yeast model of FRDA, that cells lacking FXN (Deltayfh1) had neither growth defects nor aconitase functional deficits when cultured anaerobically but, under aerobic conditions, showed a high degradation of aconitase, an accumulation of carbonylated proteins in the cytosol and the mitochondria, and a decrease in the cytosolic activity of the 20S proteasome compared to wild-type cells. Also, in the yeast model of FXN deficiency, Auchère et al. [56] described a 5-fold decrease in total glutathione (GSH + GSSG), an imbalance in GSH–GSSG pools, higher GPx activity, and a 3-fold increase in glucose-6-phosphate dehydrogenase activity (suggesting a stimulation of the pentose phosphate pathway) compared to wild-type cells. These findings suggest an important role of oxidative stress in this FRDA model.

Llorens et al. [57], using RNA interference (RNAi) in Drosophila flies to suppress FXN expression, showed in this model a shortened life span, reduced climbing abilities, and enhanced sensitivity to oxidative stress that was parallel to the situation of FRDA patients. They found an important decrease in aconitase activity and an impairment of mitochondrial respiration (suggesting an important role of oxidative stress), while activities of mitochondrial respiratory chain complexes I to IV were normal [57]. The same group described increased iron, zinc, copper, manganese, and aluminum levels [58], as well as increased levels of lipid peroxidation markers, such as MDA + 4-hydroxyalkenals (HAE) [58,59], and of GSSG [59] in this model compared to control animals [58]. In the same model, Navarro et al. [60] described the accumulation of lipid-like droplets composed of high levels of free fatty acids and a marked increase of lipoperoxides in glial cells, which were related to a shortening in lifespan.

Rodríguez et al. [61] showed, in cultures from the human SH-SY5Y neuroblastoma cell line and in a RNAi Drosophila fly model of FRDA, a compromise of the function of Endoplasmic Reticulum mitochondria-associated membranes (MAMs, involved in the regulation of essential cellular processes such as lipid metabolism and calcium signaling), which improved with treatment with antioxidants and the promotion of mitochondrial calcium uptake.

Shidhara et al. [62], in Drosophila larvae with reduced FXN expression (DfhIR), showed a decrease in mitochondrial membrane potential in cell bodies, axons, and neuromuscular junctions (NMJs) of segmental nerves, as well as defects in axonal transport in mitochondria without a basal increase in reactive oxygen substances in any neuronal region, although this was notably increased after exposure to antimycin A.

Calmels et al. [63] developed a murine cellular model of FRDA using a combination of cell lines carrying a FXN conditional allele with an EGFP-Cre recombinase to create murine cellular models depleted for endogenous FXN and expressing missense-mutated human FXN. In this model, the absence of FXN was lethal, but cells expressing the mutated FXN showed affected mitochondrial, cytosolic, and nuclear iron–sulfur cluster enzyme activities; mitochondrial iron accumulation; and an increased sensitivity to oxidative stress (similarly to FRDA patients) that were correlated with disease severity.

Wagner et al. [64], using two conditional mouse models of FRDA (neuron-specific enolase NSE and muscle creatine kinase MCK–Cre knock-out mice), characterized by the development of a fatal cardiomyopathy and impaired activity of iron–sulfur cluster-dependent respiratory complexes, showed a marked hyperacetylation of many proteins that was related to the inhibition of the NAD(+)-dependent sirtuin 3 (SIRT3) deacetylase and caused by an important decrease in the mitochondrial NAD(+)–NADH ratio and direct carbonyl group modification of SIRT3, which was reversed by incubating with excess SIRT3 and NAD+.

Sanz-Alcázar et al. [65] showed that FXN deficiency in primary cultures of dorsal root ganglia neurons and dorsal root ganglia from the FXNI151F mouse model and the decreased enzymatic activity of mitochondrial respiratory chain complexes (specifically, I and II) caused alterations in mitochondrial morphology, decreased the NAD+–NADH ratio and SIRT3 activity, increased the acetylation of alpha-tubulin and SOD2 (leading to their inactivation), and increased mitochondrial levels of superoxide anion, Fe2+, and 4-HNE. In the same models, this group described the upregulation of transferrin receptor 1 and decreased ferritin and the impaired activation of NRF2 (which caused the downregulation of SLC7A11—important for glutathione synthesis—and GPx4—increasing increased lipid peroxidation), which seemed to be due to an increase in Kelch-like ECH-associated protein 1 (KEAP1), the activation of Glycogen synthase kinase-3β (GSK3β), and deficiency in the Liver Kinase B1–AMP-activated protein kinase (LKB1–AMPK) pathway (involving the reduction of levels of the kinase LKB1 and pAMPK), as well as a decrease of SIRT1 (activator of LKB1), suggesting the presence of ferroptosis [66].

In the MCK–Cre model, increased iron levels have been shown in the heart, liver, spleen, and kidneys (an increase of iron in the liver, spleen, and kidneys seemed to be a “protective” mechanism to increase systemic iron levels and iron loading in tissues where FXN expression was intact), related to mitochondrial aggregates of iron, phosphorus, and sulfur that likely contribute to oxidative stress, as well as iron supplementation, which improved cardiac hypertrophy without affecting FXN levels [67].

Shan et al. [68], in a YG8R hemizygous mouse model of FRDA (which showed defects in movement and growth defects in dorsal root ganglia neurites) using a microarray and q-RT-PCR, showed an important decrease in transcripts encoding antioxidants (peroxiredoxins, glutaredoxins, and GST), a significant decrease of Nrf2 in the cerebellum and dorsal root ganglia that was correlated with FXN expression, and a reduction in total GSH levels. Sandi et al. [69] showed, in fibroblast and neural stem cells from YG8R mice (compared to control Y47R mouse cells), decreased FXN expression, increased DNA methylation, increased sensitivity to oxidative stress, decreased aconitase activity, the downregulation of peroxisome proliferator-activated receptor gamma coactivator 1-alpha (PGC-1α) and antioxidant gene expression levels (especially SOD2 and GPx1), and a significant reduction in the expression of several DNA mismatch repair (MMR) genes.

Abeti et al. [70,71], in cultured fibroblasts of knock-in–knock-out (KIKO; B6.Cg-*Fxntm1MknFxntm1Pand*/J) and YG8R FRDA mouse models, compared with their appropriate controls (wild-type and Y47R), showed increased lipid peroxidation and decreased mitochondrial membrane potential (the latter due to an inhibition of complex I, which is partially compensated for by an overactivation of complex II).

Codazzi et al. [72], in induced pluripotent stem cell (iPSC)-derived neurons from FRDA patients, when compared to those of controls, showed a decrease in iron–sulfur (Fe–S) and lipoic acid-containing protein concentrations; an increase in labile iron pool (LIP), ROS, and SOD2 expression; and a decrease in GSH concentrations.

Igoillo-Esteve et al. [73] showed that in pancreatic β-cell death in cultures of the clonal rat insulinoma cell line (INS-1E), primary rat β-cells, dispersed rat and human islets, and iPSC-derived neurons from FRDA patients, FXN deficiency was a consequence of the oxidative stress-mediated activation of the intrinsic pathway of apoptosis with the participation of the pro-apoptotic Bcl-2 family members Bad, DP5, and Bim.

Moreno-Lorite et al. [74], in a human neural cell line with doxycycline-induced FXN knockdown derived from the human neuroblastoma SH-SY5Y (iFKD-SY), which can differentiate into mature neuron-like cells, showed that the induction of FXN deficiency was accompanied by increases in oxidative stress, DNA oxidative damage (with concomitant transcriptional deregulation in many of the genes implicated in DNA repair pathways), decreased aconitase enzyme activity, increased levels of p53 and p21, the activation of caspase-3, and apoptosis.

Hackett et al. [75] showed that in yeast plasmids, the Lon protease Pim1 is responsible for FXN degradation and the increased turnover of Yfh1 under situations of mitochondrial oxidative stress. They also showed decreased ferritin levels in cultures of fibroblasts from FRDA patients (compared to those of controls) incubated in media with elevated iron and in cultures of rat H9C2 cardiomyocytes treated with doxorubicin.

Interestingly, Navarro et al. [76], in transgenic Drosophila flies that overexpressed human or fly FXN, showed the presence of oxidative stress (including a reduction of the level of aconitase and a decrease in the level of subunit S3 of the complex I of the mitochondrial respiratory chain), the impairment of the normal embryonic development of muscle and the peripheral nervous system, a reduction of life span, and impairment in locomotor ability and brain degeneration. Edenharter et al. [77], also in a Drosophila model, described that overexpression of FXN led to an increase in oxidative phosphorylation, a modification of mitochondrial morphology, an alteration in iron homeostasis, and the triggering of oxidative stress-dependent cell death, with cell survival being improved with genetic manipulation of mitochondrial iron metabolism by silencing mitoferrin. Similarly, Vannocci et al. [78] showed, in a human cellular model with an overexpression of FXN (HEK-cFXN, a biallelic knock-out of the endogenous *FXN* gene with the presence of an exogenous, inducible cDNA *FXN* cassette), a significant increase of oxidative stress and LIP levels, mitochondrial dysfunction, and decreased ATP production, which was similar to that found in FXN deficiency. Moreover, the transfection of the FXN gene into lymphoblasts of FRDA compound heterozygotes (FRDA-CH) with deficient FXN expression (which show high sensitivity to iron and hydrogen peroxide-induced oxidative stress) rescued fully or partially the increase in mitochondrial iron levels, the decrease in aconitase and isocitrate dehydrogenase (enzymes related to the support of mitochondrial membrane potential), and the decrease in mitochondrial membrane potential [79].

## 5. Data from Omics (Proteomics, Metabolomics) Studies

### 5.1. Omics Studies in Patients with FRDA Compared to HC

A proteomic study performed on plasma from 42 FRDA patients and 20 age- and sex-matched HC showed 13 differentially expressed proteins (DEPs) between the two groups, with albumin (considered as an oxidative stress marker) being upregulated in FRDA patients [15]. Another proteomic study in PBMC from 25 FRDA patients and 24 age- and sex-matched HC showed eight DEPs related to neuroinflammation, cardiomyopathy, altered glucose metabolism, and iron transport, and among them, decreased pyruvate dehydrogenase subunit E1 (PDHE1) and increased serotransferrin [80].

A proteomic analysis of cultured B-lymphocytes from a patient diagnosed with FRDA and his asymptomatic brother showed 278 DEPs, and among them, a significant decrease in FXN; mitochondrial respiratory chain complexes I, II, and III; and enzymes involved in mitochondrial metabolism and in defense against oxidative stress [81].

Napierala et al. [82], using a new proteomic technique—reverse-phase protein array—in 44 samples of fibroblasts from FRDA patients and 18 HC, found 30 DEPs (20 upregulated and 10 downregulated), including decreased FXN and the increased expression of the enzymes aldehyde dehydrogenase 1 family member 3 (ALDH1A3) and aldehyde-oxidase 1 (AOX1), associated with oxidative stress and the regulation of retinoic acid levels.

Indelicato et al. [83], in a proteomic analysis in skeletal muscle of five FRDA patients and four age-matched HC, identified 228 DEPs (227 downregulated), with most of them related to oxidative phosphorylation, ribosomal elements, mitochondrial architecture control, and fission–fusion pathways and 74% of them being targeted to NRF2.

A proteomic study performed in the cerebrospinal fluid (CSF) of five FRDA patients and nineteen controls found thirty-four DEPs (twenty-eight upregulated and six downregulated) with a high significance between the two study groups. Most of these DEPs were related to markers of neuroinflammation and neurodegeneration, with GPX3 (overexpressed) being the only one related to oxidative stress [84].

O’Connell et al. [85] performed a metabolomic analysis of plasma from 10 FRDA patients and 11 age- and sex-matched HC, including 540 metabolites, using a combination of two techniques and found 59 metabolites that were significantly different between the two groups, mainly associated with one-carbon (1C) metabolism, composed of formate, sarcosine, hypoxanthine (a precursor of uric acid, related to oxidative stress), and homocysteine.

In a metabolomic and lipidomic analysis of fibroblasts from nine FRDA patients and nine age- and sex-matched HC by Wang et al. [86], the main findings were an increase in 3-hydroxy-3-methylglutaryl-coenzyme A (HMG-CoA) and β-hydroxybutyrate-CoA levels, increased levels of several ceramides (especially long-chain fatty acid ceramides related to oxidative stress; these were correlated with the GAA repeat length and the frataxin protein levels), and increased levels of PUFA-containing triglycerides and phosphatidylglycerols. As we discussed previously, another lipidomic analysis did not show differences in lipid peroxidation in erythrocytes from patients with FRDA compared to HC [12].

### 5.2. Omics Studies in Experimental Models of FRDA

A proteomic analysis of heart tissue from 9-week-old MCK–Cre conditional frataxin KO mice compared to wild-type mice showed a decreased expression of complexes I and II of the mitochondrial electron transport chain and enzymes involved in ATP synthesis, as well as an increased expression of enzymes involved in the citric acid cycle, pyruvate decarboxylation, and oxidative stress protection [87].

Purroy et al. [88], in a proteomic analysis performed for a cardiac cellular model of FRDA based on neonatal rat cardiac myocytes and lentivirus-mediated frataxin RNA interference, described decreased content of the PDHA1 subunit from pyruvate dehydrogenase. The same group, using selected reaction monitoring-based targeted proteomics (including 21 proteins related to oxidative stress) in mice carrying the FXN I151F mutation and wild-type mice, showed a marked decrease in mitochondrial aconitase (ACO2) and the two components of the oxidative phosphorylation (OXPHOS) complex II in the cerebrum and cerebellum at 21 and 39 weeks, as well as for complex II at 21 weeks in the heart, as the most relevant changes, although SOD2 was also overexpressed in the cerebrum at 21 and 39 weeks and underexpressed in the heart at 21 weeks and overexpressed at 39 weeks [89].

A target metabolomic analysis of the cerebellum of the KIKO mouse model of FRDA showed a shift towards glycolysis and the production of itaconate (related to the Nrf2 and GSH pathway) and, therefore, to oxidative stress in microglia [90].

Finally, a multi-omic (metabolomic and transcriptomic) analysis of cardiac mitochondrial stress in three mouse models with different degrees of frataxin deficiency (YG8-800, KIKO-700, and FXN^G127V^) showed an increase in metabolites related to glutathione and purine metabolism, the citric acid cycle (tricarboxylic acid cycle), and several amino acid-related metabolic pathways in female KIKO-700 hearts, as well as changes in only six metabolites (4-pyridoxic acid, NADH, kynurenic acid, glutathione, deoxyribose 5-phosphate, and guanosine triphosphate) in YG8-800 hearts. However, differentially expressed genes consistent with cardiomyopathy and mitochondria-integrated stress responses were only identified in FXN^G127V^ hearts [91].

## 6. Antioxidant Therapies Tested in FRDA Patients or in Experimental Models of FRDAs

Appendix A summarizes the design and main results of clinical trials of different antioxidant drugs tested in FRDA patients [10,12,13,36,92,93,94,95,96,97,98,99,100,101,102,103,104,105,106,107,108,109,110,111,112,113,114,115,116,117,118,119,120].

### 6.1. Idebenone Alone or Associated with Deferiprone

Idebenone is a quinone derivative and synthetic analog of CoQ_10_ that acts as a transporter of the mitochondrial respiratory chain, increasing the production of ATP. Moreover, this compound is a potent free-radical scavenger and has important effects against lipoperoxidation. To date, idebenone is the drug for which the highest number of clinical trials have been conducted, including randomized clinical trials, in patients diagnosed with FRDA. Idebenone is well-tolerated even at high doses, and its administration increases plasma levels of the substance in a dose-proportional manner [121,122], although this linearity can be lost at the highest doses [122].

The pyridone derivative deferiprone (3-hydroxy-1,2-dimethylpyridin-4(1H)-one) is a potent iron chelator. To date, three clinical trials addressed its possible role in the therapy of FRDA—all of them in association with idebenone [108,109,110]—and, in one of them, in association with riboflavin as well [110].

#### 6.1.1. Studies in Patients with FRDA

Rustin et al. [92] reported for the first time a significant improvement in several echocardiographic parameters in three patients with FRDA who were administered low doses of idebenone without observing remarkable changes in neurological evaluation. These changes were accompanied by the demonstration of the improved activity of mitochondrial complex II and reduced lipid peroxidation in endomyocardial tissue samples obtained from the same patients [92]. Furthermore, another later study showed a decrease in urinary concentrations of 8-OHdG after administration of idebenone at low doses in patients with FRDA [13], suggesting a possible action of idebenone on a marker of oxidative stress in patients with that disease. However, other authors did not find any change in this parameter in patients treated with different doses of idebenone in comparison to a placebo [100].

Since then, results from several clinical trials have been published, including seven open-label studies [94,95,96,98,99,101,102], one placebo-controlled crossover study [93], and six double-blind placebo-controlled studies [97,100,104,105,106,107], whose results are summarized in Appendix A. In most of these studies, no significant improvement was shown in the scores of the clinical scales used to evaluate neurological deterioration [93,97,101,104,107], and some even describe the worsening of these scales [98,99,103]. Some have reported improvement in these clinical scales [96], at least in patients treated with intermediate or high doses of idebenone [100,106], and others only showed improvement in speech capability tests [107].

Regarding the effect of idebenone on echocardiographic or neuroimaging parameters, most studies showed improvement in some of them (the specific details are summarized in Appendix A) [20,94,95,97,98,99], while others did not find significant changes [93,96,101,102,105].

Two systematic reviews of the literature concluded that idebenone had a positive effect on cardiac hypertrophy and had a non-significant effect on neurological function in adults [123,124] but can induce a modest improvement in pediatric patients with FRDA [123].

The possible efficacy of idebenone in association with deferiprone in the treatment of FRDA has been tested in two open-label studies [108,110] (one of them used riboflavin, as well [110]) and in a randomized double-blind placebo-controlled study [109] (Appendix A). These three studies showed improvement in several echocardiographic parameters. However, regarding neurological function, one of them described improvement in three out of five patients [109], another described a lack of improvement [108], and another had a better annual worsening rate compared with an untreated FRDA cohort [110]. Velasco-Sánchez et al. [108] described improvement in iron deposits in the dentate nucleus, assessed using MRI.

#### 6.1.2. Studies in Human Cell Cultures

García-Giménez et al. [33] described that idebenone was able to inhibit mitochondriogenic responses in skin fibroblast cultures from three patients with FRDA by reversing the increased expression of PGC-1α. Similarly, idebenone has shown the capacity of preventing cell death due to endogenous oxidative stress (related to the block of glutathione synthesis) in cultured fibroblasts from FRDA patients [31,125,126,127].

Quesada et al. [47] reported that cocultures of periodontal ligament cells from three FRDA patients with idebenone and deferiprone decreased caspase expression and increased antioxidant gene and FXN expression.

Lee et al. [128], in a human iPSCl (hiPSC)-derived cardiomyocyte model from a skin biopsy of a FRDA patient, showed improvement in ROS production (related to reduced iron accumulation) and in the decay velocity of calcium-handling kinetics with treatment with deferiprone, but not with idebenone administration.

#### 6.1.3. Studies in Experimental Models of FRDA

Seznec et al. [129], in a FXN-deficient mouse model of FRDA with isolated cardiac disease confirmed by echocardiographic, biochemical, and histological studies, showed that the administration of idebenone (and not of a placebo) was able to delay cardiac disease onset, progression, and death.

Idebenone prevented oxidative stress and reduced lipid peroxidation in two cellular models of FRDA (a yeast strain lacking the *yeast FXN homolog Yfh1* gene and, in the murine fibroblast, I154F missense-mutation model) [130].

Soriano et al., in an experimental model of FRDA in knockdown flies with 30% FXN, described improvement in life span and in motor abilities after the administration of idebenone [131] (which caused the recovery of the reduction of aconitase activity) or deferiprone [58,131] (which was related to a reduction of labile iron in the mitochondria) and with copper and zinc chelators [58] without inducing changes in SOD1 activity [58].

### 6.2. Coenzyme Q_10_, Vitamin E, and Vitamin E Derivatives

Coenzyme Q_10_ is a powerful antioxidant that plays an important role in mitochondrial oxidative phosphorylation. Vitamin E, which exists in eight forms—four tocopherols and four tocotrienols—also has an important role as an antioxidant. The results of clinical trials with coenzyme Q_10_, vitamin E, and vitamin E derivatives are summarized in Appendix A.

Three clinical trials, two of them open-label [37,111] and another double-blind and placebo-controlled [19], addressed the effect of the coadministration of CoQ_10_ and vitamin E in patients with FRDA (Appendix A). None of these studies showed significant improvement in echocardiographic measures or global clinical scales of ataxia, although two reported a modest benefit in posture and gait subscales [37,111]. Interestingly, an increase in the cardiac and skeletal muscle phosphocreatine–ATP ratio and ATP synthesis induced by this treatment was demonstrated by ^31^P-MRS [37,111].

One short-term double-blind placebo-controlled study with the vitamin derivative alpha-tocopheryl–quinone showed improvement in the Friedreich’s Ataxia Rating Scale (FARS) and in the Global Impression of Clinical Severity (CGIS) [67]. Another study showed that prolonged therapy (two years) with alpha-tocotrienol–quinone improved FARS scores when compared to a matched cohort without any treatment [113]. Finally, an open-label study involving only seven patients under therapy with idebenone failed to show any improvement with a tocotrienol mixture as add-on therapy in clinical neurological scales, echocardiographic measures, and neuroimaging studies, despite improvement in some biochemical parameters [12].

Both coenzyme Q_10_ and vitamin E were able to prevent cell death due to endogenous oxidative stress (related to the block of glutathione synthesis) in cultured fibroblasts from FRDA patients [125]. Petrillo et al. [31] described that the vitamin E analog alpha-tocotrienolquinone (as well as idebenone) increased mRNA FXN levels and reversed the increased levels of endogenous lipid peroxidation and mitochondrial ROS (mROS), as well as the decreased GSH, in human skin fibroblasts from FRDA patients. The administration of tocotrienol to PBMC in FRDA patients caused a significant increase in FXN-3 (but not in FXNs 1 and 2) isoform expression, which seemed to be independent of Peroxisome Proliferator-Activated Receptor Gamma (PPARG) expression [132]. In addition, MitoVit E (a vitamin E analog that is taken up and accumulated into mitochondria) reduced ROS production in a yeast FXN knock-out model of FRDA [133].

### 6.3. Recombinant Human Erythropoietin

Boesch et al. [114], in an open-label study, showed significant improvement in neurological clinical scales and in biochemical parameters (serum FXN, peroxide, and urinary 8-OHdG levels) in a short series of patients with FRDA treated with recombinant human erythropoietin. However, these results were not confirmed in two further double-blind placebo-controlled trials [115,116].

### 6.4. Resveratrol

Resveratrol, produced by several plants, is a stilbenoid derivative, a type of natural phenol, and a phytoalexin with important antioxidant actions. An open-label study involving 24 FRDA patients with low or high doses of resveratrol showed improvement in FARS, International Cooperative Ataxia Rating Scale (ICARS), and pitch perception, but not in quality-of-life scales or echocardiographic parameters [117]. This study showed a decrease in plasma F2-isoprostane levels with a high dose of resveratrol, but not in PBMC FXN or in urinary 8-OHdG concentrations [117].

Resveratrol increased *FXN* gene expression and FXN levels in lymphoblast and fibroblast cell lines derived from individuals with FRDA and in a humanized GAA repeat expansion mouse model of FRDA [134], but not in pluripotent stem cell-derived neurons in FRDA patients [135].

### 6.5. Omaveloxolone

Omaveloxolone is a synthetic oleanane triterpenoid compound that can activate the antioxidative transcription factor (Nrf2) and inhibit the pro-inflammatory transcription factor NF-κB-light-chain-enhancer of activated B cells. Omaveloxolone improved FARS and the scores of a quality-of-life scale in a double-blind, randomized, placebo-controlled, parallel-group trial involving 103 adult FRDA patients [118]. This drug was indicated for patients with FRDA.

Moreover, omaveloxolone restored mitochondrial complex I deficiency in cerebellar granular cells in two FRDA mouse models (KIKO and YG8R) [136] and in human fibroblasts in FRDA patients [31,136]. It also increased mRNA FXN levels [31] and reversed the increased level of endogenous lipid peroxidation and mROS in the former model and the decreased GSH in the latter [31,136]. A recent study showed that omaveloxolone (but not other Nrf2 activators, such as dimethyl fumarate) was able to improve cardiomyopathy in the conditional Fxn^flox/null^::MCK–Cre knock-out (FXN-cKO) mouse model of FRDA [137].

### 6.6. Other Drugs Tested in FRDA Patients

The results of clinical trials with other drugs tested in FRDA patients are summarized in Appendix A.

Treatment with (+)-epicatechin, a drug that induces mitochondrial biogenesis and antioxidant metabolism in muscle fibers and neurons, showed improvement in cardiac function, a non-significant reduction in FARS and modified FARS (mFARS) scores, and upregulation of a muscle-regeneration biomarker in an open-label study involving 10 FRDA patients [119].

An open-label pilot study involving seven patients with FRDA showed that recombinant human granulocyte-colony stimulating factor (G-CSF) was able to improve many biochemical parameters. However, it did not investigate the effect of that drug on the clinical parameters of that disease [120].

### 6.7. Other Drugs Tested in Experimental Models of FRDA

17β-estradiol (E2) [138] and other estrogen-like compounds, such as phenolic estrogens (which are independent of estrogen receptors) [139], the soy-derived estrogen receptor β agonist S-equol, and its estrogen receptor α-preferring enantiomer, R-equol [140], have shown a cytoprotective effect in FRDA skin fibroblasts from induced oxidative insults, causing the inhibition of de novo glutathione (GSH) synthesis.

Marmolino et al. [35] described that treatment with a PGC-1alpha activator and PPARG agonist (Pioglitazone) or with a cAMP-dependent protein kinase (AMPK) agonist (AICAR) restored PGC-1alpha levels—and Mn-SOD decreased mRNA expression and protein levels—in skin fibroblasts in FRDA patients and in the cerebellum and spinal cord of a mouse KIKO model of FRDA. Rodríguez-Pascau et al. [141] showed that treatment with leriglitazone (another PGC-1alpha activator and PPARG agonist) increased FXN protein levels, reduced neurite degeneration and α-fodrin cleavage mediated by calpain and caspase 3, improved mitochondrial functions and calcium homeostasis, and increased survival in FXN-deficient dorsal root ganglia neurons. Further, it prevented lipid droplet accumulation in FXN-deficient primary neonatal cardiomyocytes and improved a motor function deficit in the YG8sR FRDA mouse model.

Marobbio et al. [133] showed a decrease in ROS production in a yeast FXN knock-out model of FRDA with pretreatment with N-acetylcysteine or with the inhibitor of TOR kinases, rapamycin. The latter was also able to produce a decrease in the mitochondrial mass in mutant cells in this model. Calap-Quintana et al. [59] showed in a Drosophila RNAi-based model of FRDA that a genetic reduction in TOR complex 1 (TORC1) signaling or pharmacologic inhibition of TORC1 signaling by rapamycin (which increased the transcription of antioxidant genes, including Nrf2 as well) improved the impaired motor performance phenotype of FRDA model flies and reversed the increased lipid peroxidation markers MDA and 4-HAE and the increased total glutathione. Petrillo et al. [31] reported that pretreatment with N-acetylcysteine, sulforaphane, or dimethyl fumarate increased mRNA FXN levels and reversed the increased level of endogenous lipid peroxidation and mROS, as well as the decrease of GSH in human skin fibroblasts in FRDA patients.

N-acetylcysteine improved the survival, locomotor function, resistance to oxidative stress, and aconitase activity in a Drosophila model of FRDA with CRISPR/Cas9 insertion of approximately 200 GAA in the intron of the fly *FXN* gene fh [142]. Despite N-acetylcysteine being able to increase GSH levels and attenuate the induction of the NRF2, KEAP1, and BTB domain and CNC homolog 1 (BACH1) induced by sulforaphane in cultures from the human SK-N-MC neuroepithelioma cell type, its administration to MCK–Cre conditional FXN knock-out (MCK KO) mouse models did not lead to improvement in the cardiac hypertrophy suffered by these animals [143]. The administration of N-acetylcysteine or the pyruvate dehydrogenase cofactors thiamine and lipoic acid to cultures of FXN-deficient neonatal rat cardiac myocytes did not prevent accumulation of lipid droplets in these cells, while the pyruvate dehydrogenase activator dichloroacetate and the superoxide-scavenger mitochondria-targeted Tiron did have a preventive effect [89].

Jones et al. [144] described that the addition of adipose stem cells from healthy humans to cultures of periodontal ligament cells from FRDA patients had a protective effect on oxidative stress (increasing cell survival) on them, which was related to the upregulation of oxidative stress-related genes and the *FXN* gene and the expression of several—especially brain-derived—neurotrophic factors (BDNFs). Similarly, treatment of fibroblasts derived from FRDA patients [34] or the siRNA-induced knockdown of FXN in SH-SY5Y cells (an experimental model of FRDA) [145] with conditioned bone marrow-derived mesenchymal stem cells (MSCs) decreases oxidative stress by restoring the decreased expression of SOD1 and SOD2 and the FXN deficiency.

The addition of mesenchymal stem cells from YG8 mice (a model of FRDA) or wild-type mice to cultures of dorsal root ganglia primary cultures isolated from YG8 mice exposed to oxidative stress resulted in the increased survival of these cells, which was related to an increase in oxidative stress-related markers and FXN expression levels, BDNF, neurotrophin 3 (NT3), and neurotrophin 4 (NT4) trophic factors [146].

Cotticelli et al. [130] described that treatment with the polyunsaturated fatty acids (PUFA) linoleic and α-linolenic, deuterated at peroxidation-prone bis-allylic positions, was able to rescue oxidative stress and reduce lipid peroxidation in a yeast strain lacking the *Yfh1* gene and murine fibroblast in a I154F missense-mutation model in a similar way to treatment with idebenone.

Methylene blue, which acts as an alternative electron carrier that bypasses mitochondrial complexes I-III, shows the ability to prevent heart dysfunction in a Drosophila model of FRDA [147]. Moreover, some methylene blue analogs increased FXN levels and mitochondrial biogenesis, improved aconitase activity, acted as reactive oxygen scavengers, inhibited complex I and glutathione depletion, preserved mitochondrial membrane potential, and increased ATP production, resulting in an important neuroprotective effect in cultured lymphocytes from FRDA patients [148].

Liver growth factor (LGF), administered intraperitoneally to transgenic mice of the FXNtm1MknTg (FXN)YG8Pook strain, restored motor coordination and showed a neuroprotective effect on neurons of the lumbar spinal cord and improved cardiac hypertrophy, with these events being related to an increase in FXN expression in the spinal cord and heart, increased mitochondrial chain complex IV expression in the spinal cord, and the reduction of the GSSG–GSH ratio in skeletal muscle [149].

The administration of phosphodiesterase inhibitors to primary cultures of sensory neurons from dorsal root ganglia of the YG8R FRDA mouse model was able to restore improper cytosolic Ca^2+^ levels and revert the axonal dystrophy found in this model [150].

Nicotinamide increased FXN gene expression and FXN levels in lymphoblast and fibroblast cell lines derived from individuals with FRDA but failed to do so in pluripotent stem cell-derived neurons from FRDA patients [135].

Villa et al. [151] showed that the systemic delivery of the catalyst activity of gold cluster superstructures (Au8-pXs) was able to improve the cell response to mitochondrial ROS and FRDA-related pathology in mesenchymal stem cells from patients with FRDA and ameliorate motor function and cardiac contractility in a YG8sR mouse model of FRDA.

A derivative of curcumin (Cur@SF NPs), which acts as a Nrf2 activator and iron chelator, has shown the ability to improve FRDA manifestation in cultures of lymphoblasts from FRDA patients and after systemic administration in a YG8R mouse FRDA model (150 mg/kg/5 days), which was related to the removal of iron from the heart, a decrease in oxidative stress, the improvement of mitochondrial function, and the compensation of FXN deficiency [152]. Abeti et al. [70] showed that other Nrf2 activators, such as sulforaphane and tricyclic bis (cyanoenone, TBE-31), protected kidney fibroblasts in KIKO and YG8R mouse models in cultures from lipid peroxidation and decreased mitochondrial membrane potential. Sulforaphane has shown the ability to improve the viability of FRDA sensory neurons generated from FRDA patient-induced pluripotent stem cells up to 61% versus the untreated control and also caused an increase of the GSH–GSSG ratio and the expression of FXN and redox markers, while omaveloxolone and dimethyl fumarate had only modest effects [153].

The administration of adenosine to a skin fibroblast culture from an FRDA patient showed a protective effect against oxidative stress and mitochondrial dysfunction (increasing cell viability) caused by an inhibitor of de novo glutathione (GSH) synthesis, which was related to the promotion of ATP production and mitochondrial biogenesis, as well as the modulation of the expression of several key metabolic genes [154].

Luffarelli et al. [155] showed a reduction of the sensitivity to hydrogen peroxide-induced cell death in FRDA fibroblasts after treatment with interferon-γ (IFN-γ), which was related to the induction of a rapid expression of Nrf2 and SOD2, leading to a potentiation of antioxidant responses.

Zhao et al. [156] showed a beneficial effect of treatment of lymphoblasts and fibroblasts derived from FRDA patients with the mitochondrion-targeted peptide SS-31, which consisted of a dose-dependent upregulation of FXN protein concentration, an increase in activities of iron–sulfur enzymes (including aconitase and complexes II and III of the respiratory chain), and improvement of mitochondrial membrane potential, ATP concentrations, and the NAD+–NADH ratio, as well as the morphology of mitochondria.

Selenium and small organoselenium compounds that act as mimics of GPx, such as ebselen, can prevent cell death due to endogenous oxidative stress (related to the block of glutathione synthesis) in cultured fibroblasts from FRDA patients [125]. In the same model, Hackett et al. [75] showed that the antihypertensive 4-hydroxychalcone and dibenzoylmethane were able to increase mRNA levels of NRF2 and FXN levels and to downregulate the antioxidants TRX, GR, and SOD2, as well as decrease ROS production.

Igoillo-Esteve et al. [157] showed that the glucagon-like peptide-1 (GLP-1) exenatide was able to improve glucose homeostasis (by enhancing the insulin content and secretion in pancreatic β cells), induce FXN and iron–sulfur cluster-containing proteins in β cells and the brain, and protect sensory neurons in dorsal root ganglia in a FXN-deficient KIKO mouse model. This drug also induced FXN expression, reduced oxidative stress, and improved mitochondrial function in FRDA patients’ iPSC-derived β cells and iPCS-derived sensory neurons. The administration of exenatide over 5 weeks to five FRDA patients showed a modest increase in FXN protein levels in platelets but not in PBMC, and increased mRNA FXN expression both in platelets and in PBMC.

A recent study by Edzemeay et al. [158] showed that the combined treatment of FRDA patient-derived fibroblasts and 2D sensory neurons with L-ascorbic acid, N-acetylcysteine, and dimethyl fumarate caused a significant decrease in mitochondrial and cellular ROS and an increase in aconitase–citrate synthase activity, GSH–GSSG ratios, and mitochondrial membrane potential. 

Finally, the transduction of human FRDA fibroblasts with recombinant adeno-associated viral and lentiviral vectors encoding the human FXN cDNA partially reversed FXN deficiency and the increased mitochondrial iron and sensitivity to oxidative stress [159].

## 7. Discussion and Conclusions

The results of studies of oxidative stress markers in different tissues of patients diagnosed with FRDA compared to those of HC have shown controversial or inconclusive results. This could be due to the scarcity of publications on the subject, the fact that many of them have a small sample size, and the differences in the assay methods and disease stage of the individuals involved in the studies. The same facts have also occurred with studies on the expression of genes related to oxidative stress in patients diagnosed with FRDA.

Among the findings of these studies, it is worth mentioning that there is (1) a decrease in the activity of mitochondrial respiratory chain complexes I, II, and III [8,27] and aconitase restricted to endomyocardial tissue [8,27] found in patients with FRDA, and (2) there is a description of the association between SNP and the *AGTR1* gene and FRDA, as well as its relation to the increased interventricular septal wall thickness and left ventricular mass in a case-control association study [44]. However, the sample sizes are too small, and the results should be confirmed by replication studies.

In contrast to the results obtained in biological samples from patients with FRDA, the data obtained from different experimental models of FRDA, based on FXN deficiency, are highly suggestive of the probable role of oxidative stress in this disease. However, as we discussed earlier, the results obtained in experimental models are not always able to be extrapolated to the disease in humans. In most of these models, the deficiency of FXN induces oxidative stress related to increased iron concentration, which causes damage and mitochondrial dysfunction; a decrease in mitochondrial respiratory chain complexes and mitochondrial membrane potential; a decrease in aconitase activity, such as the induction of antioxidant enzymes such as SOD1, SOD2, GPx, peroxiredoxins, glutaredoxins, and GST (among others); a decrease in GSH concentrations and the GHS–GSSG ratio; an increase in markers of lipoperoxidation and oxidation of DNA, as well as carbonyl proteins; and the decreased expression of NRF2 and downregulation of PGC-1α. Paradoxically, experimental situations of FXN overexpression can also cause oxidative stress [76,79].

Drawing from key findings in both patient studies and experimental models, Figure 1 illustrates the proposed interactions among the various mechanisms of oxidative stress contributing to FRDA pathogenesis. FXN mutation or the induction of FXN deficiency would lead to a loss of Fe–S clusters in various mitochondrial enzymes, which would increase iron. This, in turn, would cause mitochondrial dysfunction and an increase in ROS production, inducing oxidative stress, the induction of antioxidant enzymes, and a decrease in GSH, NRF2, and PGC1-alpha. Mitochondrial dysfunction would also cause glutamatergic disturbances and alterations in calcium homeostasis (which would, in turn, lead to an increase in ROS production). As a consequence of oxidative stress, there would be an increase in the oxidation of lipids, proteins, and DNA, and an alteration in oxidative phosphorylation (with a decrease in ATP synthesis). All these mechanisms would ultimately lead to cell death. However, we must keep in mind that the results obtained from experimental models have significant limitations. One of the most important facts is that the heterogeneity and pathophysiological complexity of human diseases mean that the results of the experimental studies cannot be fully extrapolated to what happens in them [160,161,162].

Although idebenone, CoQ10, and vitamin E derivatives have shown benefits in experimental models, clinical trials with idebenone in FRDA patients have demonstrated only modest improvement in cardiac hypertrophy and minimal neurological benefit. These effects may be slightly enhanced when combined with deferiprone, though evidence is limited to three studies, with only one double-blind trial involving five patients. Combinations of vitamin E and CoQ_10_ and vitamin E on its own have not shown beneficial effects, while vitamin E derivatives such as alpha-tocopheryl–quinone and alpha-tocotrienol–quinone caused a modest improvement in a short series of patients with FRDA (although studies with these treatments were double-blind, randomized, and placebo-controlled, and, therefore, had a high level of evidence).

The activator of Nrf2, omaveloxolone has been the only drug that has shown a clear, although modest, improvement of neurological symptoms in patients with FRDA in a double-blind, placebo-controlled trial involving a significant number of patients (and, therefore, has a high level of evidence) [118], and as of today, it is indicated as a treatment for this disease. In addition, this drug showed improvement in oxidative stress parameters [31,136] and in cardiac hypertrophy [138] in experimental models.

Resveratrol [118] and epicatechin [119] showed a modest improvement of certain clinical symptoms and G-CSF [120] of several biochemical parameters in a short series of FRDA patients (although the clinical efficacy was not assessed), but the studies with these drugs were short-term, open-label, and involved a low number of patients, so they had a low level of evidence. The promising beneficial effects of treatment with recombinant human erythropoietin in an open-label study [114] were not confirmed in double-blind placebo-controlled trials [115,116].

Many drugs have demonstrated significant benefits in reducing oxidative stress parameters in various experimental models of FRDA. These include, among others, estrogen derivatives [138,139,140], PGC-1alpha activators acting as PPARG agonists [35,141], AMPK agonists [35], TOR kinase inhibitors [59,133], N-acetylcysteine [31,133,142], sulforaphane [31,143], dimethyl fumarate [31], PUFAs [130], methylene blue [147,148], LGF [149], phosphodiesterase inhibitors [150], NRF2 activators [70,152,153], adenosine [154], IFN-γ [155], and the combination of L-ascorbic acid, N-acetylcysteine, and dimethyl fumarate [158]. To our knowledge, the possible efficacy and safety of these drugs in patients with FRDA has not yet been explored, except in the case of the NRF2 activator omaveloxolene [118].

## 8. Future Directions

Further data from studies in different FRDA models suggest the possibility of an important role of oxidative stress mechanisms in this disease, which have been described with sufficient detail in a previous section. However, data on FRDA patients are insufficient due to the scarcity of studies published on this topic. It is striking that no data have been published on studies of oxidative stress markers in CSF (except for a proteomic study with a low sample size), despite its easy accessibility.

Although the data on the contribution of oxidative stress to the pathogenesis of FRDA inferred from experimental models are available, it would be desirable to increase the number of studies and the sample size regarding oxidative stress markers in patients with FRDA compared to HC. For future studies related to this topic, we propose that it would be advisable to meet the following conditions:Design prospective and multicenter studies with a long-term follow-up period.Ensure the participation of an important number of patients, both symptomatic and non-symptomatic, with a genetic-molecular diagnosis of FRDA, along with a comparable number of healthy age- and sex-matched individuals.Exclude subjects (both in the FRDA and HC groups) exposed to any factors that may alter oxidative stress markers, such as medication use (antioxidant vitamins, calcium or mineral supplements, diuretics, bisphosphonates), atypical dietary habits, pregnancy, some comorbidities (undernutrition, obesity, oncologic and acute infectious diseases, and thyroid, parathyroid, kidney, or liver disorders), as well as recent trauma or surgical interventions. Some other possible confounding variables, such as diet, physical activity, alcohol consumption and tobacco smoking, exposure to toxic substances, history of arterial hypertension, diabetes mellitus, and hypercholesterolemia, should be controlled.Collect plasma/serum, blood cells, and CSF to analyze oxidative stress biomarkers at baseline and after a long-term follow-up, both in FRDA patients and HC.Conduct periodic clinical evaluations (every 3–6 months) of patients with FRDA using standardized scales for FRDA [163,164,165,166,167,168] and cardiological assessments to monitor disease severity and progression.Finally, in the event of death, brain donation from FRDA patients and HC should be encouraged to enable examination of oxidative stress markers in both groups.

Regarding the use of antioxidant treatments, as previously described, many of them have shown efficacy in experimental models of FRDA, while in clinical trials in patients with this disease, only omaveloxolone and, to a lesser extent, idebenone (at least in the treatment of cardiac hypertrophy) have shown some degree of effectiveness. To better assess the effectiveness of antioxidant therapies in FRDA, future research should include large-scale, prospective, multicenter, randomized, placebo-controlled trials testing compounds that have shown promise in experimental models. However, we must keep in mind that the results obtained in experimental models with different drugs would not always be able to be extrapolated to those expected in clinical trials with patients.

## Figures and Tables

**Figure 1 cells-14-01406-f001:**
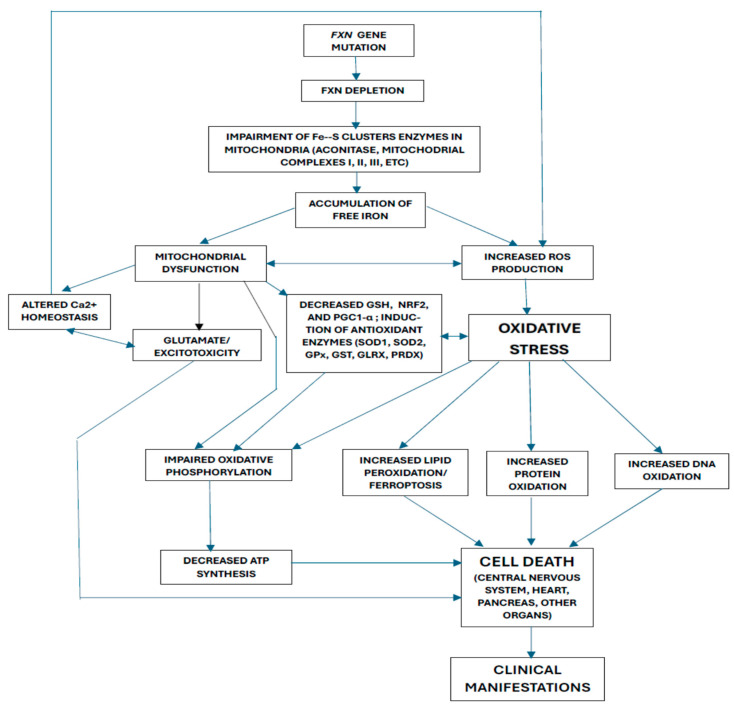
Possible interactions between the different pathogenic mechanisms proposed for Friedreich’s ataxia.

**Table 1 cells-14-01406-t001:** Summary of the results of studies addressing concentrations of oxidative stress markers in patients with FRDA compared to HC.

(1) Markers of lipid peroxidation (MDA, TBARS, 4-HNE). (1.1) Cerebellum, brainstem, and cortex: non-significant differences between FRDA patients and HC [6]. (1.2) Plasma: MDA increased six-fold [11], 4-HNE increased two-fold in fibroblasts [30], or non-significant differences between FRDA patients and HC [12]. (2) DHBA levels [13] and susceptibility to oxidative stress [11] in plasma: non-significant differences between FRDA patients and controls. (3) Superoxide levels in plasma: increased about 25% in fibroblasts from FRDA patients [33]. (4) 8-OHdG: increased by 2.6-fold in the urine of FRDA patients [13]. (5) Carbonyl proteins increased over two-fold, and TAC decreased about 30% in patients with FRDA [12]. (6) Mitochondrial respiratory chain complexes (enzymatic activities or expression). (6.1) Cerebellum: non-significant differences between FRDA patients and HC [8]. (6.2) Dorsal root ganglia: non-significant differences between FRDA patients and HC [8]. (6.3) Spinal cord: non-significant differences between FRDA patients and HC [7]. (6.4) Lymphocytes/lymphoblasts: non-significant differences between FRDA patients and HC [27]. (6.5) PBMC: non-significant differences between FRDA patients and HC [29]. (6.6) Platelets: non-significant differences between FRDA patients and HC [29]. (6.7) Skin fibroblasts: non-significant differences between FRDA patients and HC [27]. (6.8) Skeletal muscle: non-significant differences between FRDA patients and HC [8,27]. (6.9) Endomyocardial tissue: decrease in activity in complex I by 65–85%, complex II by 75–88%, and complex III by 40–80% [8,27], and non-significant differences in activity for complexes IV [8,27] and V [27] in patients with FRDA compared to HC. (7) Iron. (7.1) Cerebellum: non-significant differences between FRDA patients and HC [8,9]. (7.2) Spinal cord: non-significant differences between FRDA patients and HC [8]. (7.3) Dorsal root ganglia: non-significant differences between FRDA patients and HC [8]. (7.4) Peripheral nerves: non-significant differences between FRDA patients and HC [8]. (7.5) Erythrocytes: non-significant differences between FRDA patients and HC [23]. (7.6) Skeletal muscle: non-significant differences between FRDA patients and HC [8]. (8) Iron-related proteins/enzymes: aconitase. (8.1) Cerebellum: non-significant differences between FRDA patients and HC [8]. (8.2) Dorsal root ganglia: non-significant differences between FRDA patients and HC [8]. (8.3) Skin fibroblasts: non-significant differences between FRDA patients and HC [11,27]. (8.4) Skeletal muscle: non-significant differences between FRDA patients and HC; [8,27]. (8.5) Endomyocardial tissue: decreased six-fold in FRDA patients [8,27]. (9) Iron-related proteins/enzymes: others. (9.1) Hemoglobin: non-significant differences in erythrocytes between FRDA patients and HC [26]. (9.2) Protoporphyrin IX: non-significant differences in erythrocytes between FRDA patients and HC [26]. (9.3) Ferrochelatase: non-significant differences in erythrocytes between FRDA patients and HC [26]. (9.4) Glutathionyl–hemoglobin: increased in erythrocytes from FRDA patients by 85% [22]. (10) Alpha-tocopherol (vitamin E). (10.1) Cerebellum, brainstem, and cortex: non-significant differences between FRDA patients and HC [6]. (10.2) Serum: similar [16,17] or decreased by 36% in FRDA patients compared to HC [18,19]. (11) Other antioxidant substances. (11.1) Coenzyme Q10: decreased by 50% in serum from FRDA patients [19]. (11.2) Uric acid: increased by 17% in serum from FRDA patients [20]. (11.3) Glutathione bound to proteins: increased in the spinal cord [7] and over two-fold in fibroblasts [32] from FRDA patients. (11.4) Total glutathione: non-significant differences between FRDA patients and HC in plasma [12], whole blood [22,23], and lymphocytes [28]. Similar [23,32] or 20% decrease in fibroblasts from FRDA patients [31]. (11.5) Free glutathione: decreased by 15% in whole blood from FRDA patients [31]. (11.6) GSH/GSSG ratio: decreased by four-fold in plasma [12], about 30% in lymphocytes [28], and about three-fold in fi-broblasts [32] from FRDA patients. (12) Antioxidant enzymes: (12.1) GPx activity: increased by 20% in whole blood from FRDA patients [24], and non-significant differences between FRDA patients and HC in erythrocytes [25] and fibroblasts [33]. (12.2) GST activity: increased two-fold in erythrocytes [25]. (12.3) Total SOD activity: increased by 50% in erythrocytes [25] and slightly decreased expression of SOD in fibroblasts from FRDA patients [34] (12.4) SOD2 activity or expression: decreased by 40% [33,34] in fibroblasts from FRDA patients. (12.5) SOD1 activity or mRNA: decreased in fibroblasts from FRDA patients [34] or non-significant differences between FRDA patients and HC [33]. (12.6) CAT activity: non-significant differences between FRDA patients and HC [33]. (12.7) TRX1 and GLRX1 protein and mRNA: significantly upregulated (about 50% and more than two-fold, respectively) in fibroblasts from FRDA patients [30].

CAT, catalase; DHBA, Dihydroxybenzoic acid; FRDA, Friedreich’s ataxia; GLRX1, Glutaredoxin 1; GPx, glutathione peroxidase; GSH, glutathione; GSSG, oxidized glutathione; GST, glutathione transferase; HC, healthy controls; 4-HNE, 4-hydroxy-2-nonenal; MDA, malonyldialdehyde; OH-dG, 8-hydroxy-deoxyguanosine; SOD1, superoxide–dismutase 1; SOD2, superoxide–dismutase 2; TAC, total antioxidant capacity; TBARS, thiobarbituric acid-reactive substances; TRX1, Thioredoxin 1.

## Data Availability

No new data were created or analyzed in this study. Data sharing is not applicable to this article.

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
