# Peer review of "Oxidative Stress and Antioxidant Therapies in Friedreich’s Ataxia"

_cells, 2025, doi:10.3390/cells14181406_

Round 1
Reviewer 1 Report
Comments and Suggestions for Authors
The manuscript presents a comprehensive narrative review of oxidative stress in Friedreich’s ataxia (FRDA), but several important issues must be addressed before it can be considered for publication in a prestigious journal. The abstract is lengthy, too broad, and needs to quantify results better, emphasize uniqueness, and differentiate between patient and model data. (2) There is an overdependence on enumerating studies without integration; the review should combine evidence to create a unified mechanistic framework. (3) The introduction repeats background information found in numerous previous reviews without offering a new conceptual viewpoint. (4) Certain methodological explanations (e.g., literature search methods) lack reproducibility—details on databases besides PubMed, criteria for inclusion/exclusion, and variations in search terms should be specified. The evidence from human and animal models is imbalanced; the discussion fails to critically evaluate the translational significance or constraints of the model systems. Numerous reported results are inconsistent (e.g., vitamin E levels), but the paper does not examine possible causes for these differences (sample size, assay method, disease stage, tissue type). (7) Tables are excessively lengthy and dense, risking a loss of reader interest—condensing, grouping by biomarker type, and adding effect direction/strength would enhance clarity. (8) Certain statistical outcomes are presented lacking effect sizes, confidence intervals, or an explanation of the clinical significance of the differences. (9) The section on genetic variants presents associations without sufficient exploration of functional validation or replication status. The part regarding antioxidant treatments is mainly descriptive; it ought to evaluate trial design, dosing, duration, endpoints, and criteria for patient selection critically. The authors' findings regarding the efficacy of antioxidant therapy are speculative and overstate the available evidence; a more cautious, evidence-driven conclusion is needed. (12) An illustrative summary is absent (e.g., a pathway diagram linking frataxin deficiency to oxidative stress markers and clinical results), which would aid in integrating findings. (13) Certain sentences are awkward in their grammar; for example, “The pathogenesis of Friedreich’s ataxia (FRDA) is not well known” should be revised to “The pathogenesis of Friedreich’s ataxia (FRDA) remains poorly understood.” (14) There is an inconsistent application of tense—shifting between past and present—that requires standardization (using past tense for study findings and present for general information). (15) The application of abbreviations lacks consistency; for example, HC (healthy controls) ought to be defined initially and used uniformly, removing the necessity to repeat “healthy controls” in parentheses later. (16) Certain figure/table captions do not provide enough information to be comprehended without consulting the main text. The formatting of the references is inconsistent; some journal titles are shortened while others are presented in their entirety. The manuscript's extent and detail could exceed what is suitable for a narrative review—it's advisable to trim excessive material and transfer supplementary content to the appendices. Expressions like “it appears logical to assume” lack scientific credibility and ought to be substituted with terminology grounded in proof. (20) There is minimal conversation regarding recent discoveries from omics-based biomarkers (metabolomics, proteomics) and their possible role in elucidating the role of oxidative stress in FRDA. (21) The authors fail to sufficiently consider confounding variables in assessments of oxidative stress (e.g., diet, medication use, and comorbidities).
Comments on the Quality of English LanguageEnglish could be improved
Author Response
(1)The abstract is lengthy, too broad, and needs to quantify results better, emphasize uniqueness, and differentiate between patient and model data.
Abstract has been rewritten according to these advices.
(2) There is an overdependence on enumerating studies without integration; the review should combine evidence to create a unified mechanistic framework.
This article was designed as a narrative review, so its main goal is to list studies of interest in the development of knowledge about the pathogenesis of Friedreich's ataxia, as well as to suggest future perspectives on how to continue advancing or investigating. The integration of the results from the mentioned studies aimed at creating a unified mechanistic framework has been carried out in the section 'Discussion,' which has been notably modified and expanded in accordance with other comments. In this regard, a figure has been added to represent the possible pathogenic mechanisms related to oxidative stress in Friedreich's ataxia.
(3) The introduction repeats background information found in numerous previous reviews without offering a new conceptual viewpoint.
The introduction is a brief summary of historical data, and basic information on epidemiological, clinical, genetic, and histopathological data, which we believe are useful for readers not familiar with the FRDA before tackling the main topics to be developed.
(4) Certain methodological explanations (e.g., literature search methods) lack reproducibility—details on databases besides PubMed, criteria for inclusion/exclusion, and variations in search terms should be specified.
We have specified in the introduction that the studies included should address one of the following points:
- Studies addressing measurements of oxidative stress marker concentrations in different tissues from patients diagnosed with FRDA (including proteomic, metabolomic, lipidomic, and multi-omic studies)
- Case-control studies on the possible association of genes related to oxidative stress with the risk for FRDA
- Studies on the presence of oxidative stress in experimental models of FRDA
- Studies addressing the possible efficacy of antioxidant drugs in the treatment of FRDA, both clinical trials in patients with FRDA and in experimental models of this disease.
(5) The evidence from human and animal models is imbalanced; the discussion fails to critically evaluate the translational significance or constraints of the model systems.
As previously discussed, in line with point (2), section 'Discussion,' which has been notably modified and expanded. We also added some comments regarding the constraints of the model systems.
(6) Numerous reported results are inconsistent (e.g., vitamin E levels), but the paper does not examine possible causes for these differences (sample size, assay method, disease stage, tissue type).
Indeed, sample sizes, assay method, etc, are possible causes of the different results. We have emphasized this aspect.
(7) Tables are excessively lengthy and dense, risking a loss of reader interest—condensing, grouping by biomarker type, and adding effect direction/strength would enhance clarity.
OK, we have moved the primitive Tables to a supplemental file, and added a Table condensing the results of measurements of oxidative stress markers grouping for biomarker type.
(8) Certain statistical outcomes are presented lacking effect sizes, confidence intervals, or an explanation of the clinical significance of the differences.
OK, Table 1 now includes the extent of the significant findings where available.
(9) The section on genetic variants presents associations without sufficient exploration of functional validation or replication status.
We have added, at the end of the section, the sentence “Most of these studies, performed with different methods, needs to be confirmed by replication studies”.
(10) The part regarding antioxidant treatments is mainly descriptive; it ought to evaluate trial design, dosing, duration, endpoints, and criteria for patient selection critically.
Table 2 of the previous version (now Supplementary Table 2) contains the sections Study design (which includes trial design, dosing, duration, primary and secondary endpoints, and criteria for patient selection) and main results (outcomes) of each of the studies in patients with FRDA. The text provides a summary of the most important data related to each drug investigated in these patients, which is described in greater detail in the table. The sections related to studies on the effect of drugs in experimental models (some of which were also tested in patients with FRDA, discussed in the section dedicated to each drug) are more descriptive in nature.
(11) The authors' findings regarding the efficacy of antioxidant therapy are speculative and overstate the available evidence; a more cautious, evidence-driven conclusion is needed.
- We have modified the discussion as indicated, adding evidence-driven conclusions
(12) An illustrative summary is absent (e.g., a pathway diagram linking frataxin deficiency to oxidative stress markers and clinical results), which would aid in integrating findings.
We have added the new Figure 1 on the proposed pathogenetic mechanisms of HD and their interactions
(13) Certain sentences are awkward in their grammar; for example, “The pathogenesis of Friedreich’s ataxia (FRDA) is not well known” should be revised to “The pathogenesis of Friedreich’s ataxia (FRDA) remains poorly understood.”
OK, corrected
(14) There is an inconsistent application of tense—shifting between past and present—that requires standardization (using past tense for study findings and present for general information).
OK, revised and corrected.
(15) The application of abbreviations lacks consistency; for example, HC (healthy controls) ought to be defined initially and used uniformly, removing the necessity to repeat “healthy controls” in parentheses later.
OK, removed from the text. All abbreviations have been exhaustively revised, corrected and defined at first use. All changes are highlighted in red colour
(16) Certain figure/table captions do not provide enough information to be comprehended without consulting the main text.
We have corrected the Tables (now Supplementary Tables) captions.
(17) The formatting of the references is inconsistent; some journal titles are shortened while others are presented in their entirety.
All references have been examined and corrected when necessary
(18) The manuscript's extent and detail could exceed what is suitable for a narrative review—it's advisable to trim excessive material and transfer supplementary content to the appendices.
Tables 1 and 2 have been transferred to supplementary content.
(19) Expressions like “it appears logical to assume” lack scientific credibility and ought to be substituted with terminology grounded in proof.
We are sorry, but we have been unable to detect the expression "it appears logical to assume" through the original text. The phrase most similar to this is "it seems reasonable to think that,…" in the abstract, which has been modified.
(20) There is minimal conversation regarding recent discoveries from omics-based biomarkers (metabolomics, proteomics) and their possible role in elucidating the role of oxidative stress in FRDA.
Ok. We have introduced a new section dedicated to omics-based studies in FRDA patients and in experimental models of FRDA, with special emphasis on oxidative stress markers.
(21) The authors fail to sufficiently consider confounding variables in assessments of oxidative stress (e.g., diet, medication use, and comorbidities).
We have modified the point 3 to include potential confounding variables for our proposition for future studies con oxidative stress markers in FRDA (highlighted in red colour). The new text is as follows:
Exclude subjects (both in the FRDA and HC groups) exposed to any factors that may alter oxidative stress markers, such as medication use (antioxidant vitamins, calcium or mineral supplements, diuretics, bisphosphonates), atypical dietary habits, pregnancy, some comorbidities (undernutrition, obesity, oncologic and acute infectious diseases, or thyroid, parathyroid, kidney, or liver disorders), as well as recent trauma or surgical interventions. Some other possible confounding variables such as diet, physical activity, alcohol consumption and tobacco smoking, exposure to toxic substances, history of arterial hypertension, diabetes mellitus, and hypercholesterolemia should be controlled
Reviewer 2 Report
Comments and Suggestions for Authors
Friedreich’s ataxia (FRDA) remains a rare but devastating neurodegenerative disease, and oxidative stress is a widely recognized component of its pathophysiology. The review topic addresses both mechanistic and therapeutic perspectives, which are important for research and clinical translation.
However, there are some major concerns:
1, While the manuscript summarizes known mechanisms, it often reports findings without critically appraising their strength, reproducibility, or limitations. For example, antioxidant clinical trials are mentioned, but the lack of consistent benefit is not sufficiently explored. The authors should evaluate the robustness of key studies, identify controversies, and discuss why some therapies failed despite strong preclinical rationale.
2, Recent advances in omics approaches, novel biomarkers of oxidative stress, and gene-editing-based therapeutic strategies (e.g., CRISPR for FXN correction) are missing. The authors should expand the discussion to include current high-impact studies from 2022–2024.
3, The link between frataxin deficiency, mitochondrial iron overload, and ROS production could be further mechanistically dissected (e.g., role of ferroptosis, mitochondrial ferritin, Nrf2 pathway regulation). The authors should include recent insights into iron-dependent cell death pathways and their experimental validation in FRDA.
4, Therapeutic approaches are listed, but their clinical trial phase, patient numbers, endpoints, and statistical outcomes are not consistently stated. The authors should add a summary table of clinical trials with key efficacy and safety data.
minor defects:
1, The review would benefit from at least one schematic diagram illustrating FRDA pathogenesis, showing how frataxin deficiency leads to oxidative stress and downstream effects.
2, The authors should ensure abbreviations (e.g., FXN, ROS, ISC) are defined at first use and used consistently.
3, Some sentences are overly long and dense—shorter, active constructions would improve readability.
4, The authors should ensure references follow the journal’s style, and double-check for missing DOI links.
Author Response
MAJOR CONCERNS:
(1) While the manuscript summarizes known mechanisms, it often reports findings without critically appraising their strength, reproducibility, or limitations. For example, antioxidant clinical trials are mentioned, but the lack of consistent benefit is not sufficiently explored. The authors should evaluate the robustness of key studies, identify controversies, and discuss why some therapies failed despite strong preclinical rationale.
This article was designed as a narrative review, so its main goal is to list studies of interest in the development of knowledge about the pathogenesis of Friedreich's ataxia, as well as to suggest future perspectives on how to continue advancing or investigating. The integration of the results from the mentioned studies aimed at creating a unified mechanistic framework has been carried out in the section 'Discussion,' which has been notably modified and expanded in accordance with other comments. In this regard, a figure has been added to represent the possible pathogenic mechanisms related to oxidative stress in Friedreich's ataxia. We also added some comments regarding the constraints of the model systems
(2) Recent advances in omics approaches, novel biomarkers of oxidative stress, and gene-editing-based therapeutic strategies (e.g., CRISPR for FXN correction) are missing. The authors should expand the discussion to include current high-impact studies from 2022–2024.
Ok. We have introduced a new section dedicated to omics-based studies in FRDA patients and in experimental models of FRDA, with special emphasis on oxidative stress markers.
(3) The link between frataxin deficiency, mitochondrial iron overload, and ROS production could be further mechanistically dissected (e.g., role of ferroptosis, mitochondrial ferritin, Nrf2 pathway regulation). The authors should include recent insights into iron-dependent cell death pathways and their experimental validation in FRDA.
We have added the new Figure 1 on the proposed pathogenetic mechanisms of HD and their interactions
(4) Therapeutic approaches are listed, but their clinical trial phase, patient numbers, endpoints, and statistical outcomes are not consistently stated. The authors should add a summary table of clinical trials with key efficacy and safety data.
Table 2 of the previous version (now Supplementary Table 2) contains the sections Study design (which includes trial design, dosing, patient numbers, duration, primary and secondary endpoints, and criteria for patient selection) and main results (outcomes) of each of the studies in patients with FRDA. The text provides a summary of the most important data related to each drug investigated in these patients, which is described in greater detail in the table. We did not include safety data except in cases where side effects have been communicated. In the new Supplementary Table 2 we have added tolerance data.
MINOR DEFECTS:
(1) The review would benefit from at least one schematic diagram illustrating FRDA pathogenesis, showing how frataxin deficiency leads to oxidative stress and downstream effects.
We have added the new Figure 1 on the proposed pathogenetic mechanisms of HD and their interactions
(2) The authors should ensure abbreviations (e.g., FXN, ROS, ISC) are defined at first use and used consistently.
OK, all abbreviations have been exhaustively revised, corrected and defined at first use. All changes are highlighted in red colour
(3) Some sentences are overly long and dense—shorter, active constructions would improve readability.
OK, revised and changed.
(4) The authors should ensure references follow the journal’s style, and double-check for missing DOI links. OK. All references have been examined and corrected when necessary. References 1, 18 and 36 have not DOI associated. The other 144 references have their DOI link mentioned.
Round 2
Reviewer 2 Report
Comments and Suggestions for Authors
The authors have revised their article according to reviewer's comments. No further comments here. I suggest accepting it in the current form.